# Metabolic Performance of Mealworms and Black Soldier Fly Larvae Reared on Food and Agricultural Waste and By-Products

**DOI:** 10.3390/ani15020233

**Published:** 2025-01-16

**Authors:** Frederik Kjær Nielsen, Rasmus Juhl Hansen, Asmus Toftkær Muurmann, Simon Bahrndorff, Niels Thomas Eriksen

**Affiliations:** Department of Chemistry and Bioscience, Aalborg University, Fredrik Bajers Vej 7H, DK-9220 Aalborg, Denmark; fkniel@gmail.com (F.K.N.); rasmusjh99@outlook.dk (R.J.H.); asmustm@bio.aau.dk (A.T.M.); sba@bio.aau.dk (S.B.)

**Keywords:** *Tenebrio molitor*, *Hermetia illucens*, feed substrate, feed efficiency, net growth efficiency, cost of growth, maintenance

## Abstract

Feed efficiency is one of the most important aspects of insect production, affected not only by the quality of the feed substrates, but also how well insect species have adapted to feed substrates and their metabolic performance. Here, we compare the metabolic performance of mealworms and black soldier fly larvae, two of the most widely used industrial insects, on a selection of by-products from agricultural and food industries. Feed substrates affected growth more in mealworms than in black soldier fly larvae. With respect to feed efficiency, the black soldier fly larvae were advantageous in terms of the highest specific growth and feed assimilation rates and the shortest development period. Mealworms were advantageous in terms of the lowest maintenance rates. The mealworms converted 16–40% of assimilated feed into growth in terms of carbon equivalents, and the BSF larvae 33–56%. Thus, it seems that BSF larvae are more versatile and slightly more feed efficient than mealworms due to differences in their metabolic performance.

## 1. Introduction

Insect farming for food and feed based on agricultural waste- and by-products are young industries still under development [1]. Different insect species have adapted to different feeds, and comparative studies of insect performance under comparable conditions may help to match insects to waste- and by-products. The efficiency by which feed substrates are converted into insect biomass, the feed efficiency, is a key aspect in insect farming. It is, however, challenging to compare feed efficiency measurements across studies due to non-standardized experimental designs and the many ways in which the feed efficiency can be affected along the way while the insects are being reared. Feed efficiency is also not calculated and reported the same way in all studies. The feed efficiency is species-dependent and influenced by the origin and nutritional composition of the feed substrates, their texture and particle size distributions [2,3], moisture content [4,5], feeding regime [6], larval density [7,8,9,10], co-occurring microbial processes in the feed substrates [5,11], host microbiome [12,13], and probably more. Furthermore, the insect larvae live mixed with their feed substrates, which are gradually transformed into frass, complex matrices of feed components and fecal pellets.

Larvae of the yellow mealworm beetle, *Tenebrio molitor*, and the black soldier fly (BSF), *Hermetia illucens*, are two insects that are reared at an industrial scale for food or feed. Mealworms are typically fed dry and shallow feed substrates such as cereal products but also accept other feed substrates. Mealworms reach weights of up to around 60–100 mg dry weight (DW) at 60–120 days of age [14,15,16,17,18]. BSF larvae are fed wet feed substrates and grow faster and often bigger than mealworms, reaching weights above 100 mg DW in as little as 2 weeks [19]. BSF larvae have been successfully reared on many different feed substrates [20]. The yellow mealworm beetle and the BSF thus represent species that have adapted to different food and lifestyles, and this may affect their efficiency in converting waste and by-products differently. They also represent different taxonomic orders, Diptera and Coleoptera.

Wheat bran or chicken feed are common base line feeds in studies of mealworms and BSF larvae, while abundant waste or by-products from food, agricultural, or other sectors are more attractive for use as feed substrates at a large scale. Several different performance indicators are used to describe the feed efficiency in insects and other animals. One is the substrate conversion efficiency, SCE, which is the ratio between the increase in larval dry weight, DW, and the amount of feed substrate (in terms of dry matter) removed during the same period (some papers use alternative names or calculate the SCE based on fresh weights). SCEs (sometimes also reported as the reciprocal feed conversion ratio) have been highly variable in both species, from 0.02 to 0.73 in mealworms [14,15,16,17,18] and 0.03–0.50 in BSF larvae [20,21,22]. Not only the feed quality affects the SCE. Co-occurring microbes may, if the conditions are right, consume considerable amounts of feed substrate [5], thereby lowering the SCE independently of the performance of the larvae, or in collaboration with the insect host [13]. The larvae’s physiology and growth patterns are also important. Large insect species have lower metabolic rates relative to body weight (specific metabolic rates) than smaller species [23]. Large insects thus spend food components for respiratory purposes relatively slowly. This is at least the case across insect species, although not necessarily across different sized individuals of the same species [23]. Furthermore, high growth rates minimize the time it takes for larvae to reach their harvest weight, during which time they also invest a portion of their assimilated feed for maintenance purposes. Thus, larger body weights and higher growth rates should favor higher substrate conversion efficiencies in BSF larvae compared to mealworms.

The feed efficiency is also influenced by the metabolic performance of the insect larvae [24]. In BSF larvae reared on chicken feed, 53–58% of the assimilated feed, in terms of carbon equivalents, is retained as growth until the larvae reach the prepupal stage [22]. This has been termed as the average carbon net growth efficiency, NGE*_avg_ of the larvae [24]. NGE*_avg_ describes the ratio between the amounts of carbon stored as growth and carbon assimilated into the metabolism across the life span of the larvae. Part of the carbon assimilated is lost as CO_2_. indicator that reflects the performance of just the larvae. BSF larvae spend a quantity of metabolites daily that corresponds to 5–13% of their own carbon content on maintenance. Growth results in additional metabolic losses, the cost of growth, in the order of 40% or more of the gained biomass weight [5,24,25]. These papers assumed that the BSF larvae and their feed had similar carbon contents. If so, the NGE*_avg_ attains the same value as the overall net growth efficiencies based on the energy or mass balances of the larvae. Different feed substrates lead to different maintenance coefficients and costs of growth, ultimately impacting the NGE*_avg_. The metabolic performance of mealworms has not yet been characterized in the same detail.

In this study, we characterize and compare the metabolic performance, in terms of the specific rates of feed assimilation, growth, respiration, maintenance, and cost of growth of mealworms and BSF larvae reared on feed substrates of variable quality, made from agricultural or food by- or waste-products. The purpose is to elucidate the relationship between metabolic performance and feed efficiency in the two species in order to explain potential species-specific differences in feed efficiency and assess and compare their potential for converting diverse feed substrates into own biomass.

## 2. Materials and Methods

### 2.1. Mealworms and BSF Larvae

Mealworms were produced from eggs prior to the rearing experiments. Ca. 400 adult mealworm beetles were obtained from the Danish company, Bugging Denmark, Copenhagen, Denmark. The beetles were placed in a rectangular tray (56 cm × 36 cm, height = 11 cm) with slices of carrot and 0.5 kg of wheat flour to facilitate oviposition. The tray was placed in a dark climate chamber at 28 °C and a humidity of 85–90%. Eggs were sieved (mesh size 500 nm) from the flour every three days and placed in cylindrical containers with a diameter of 7.5 cm and a height of 11.5 cm (nurseries) with wheat bran in a dark climate chamber at 28 °C, 85–90% humidity. Neonate larvae were observed after 10–12 days, and after another 10 days (21 days in total), the larvae had reached a size where they could be handled without causing them damage and were transferred to the experimental diets.

BSF larvae, 7 days of age, were obtained from the Danish insect producing company, Enorm Biofactory, Flemming, Denmark. The larvae were transferred directly from their original grain-based feed to the respective feed substrates used in the rearing experiments.

### 2.2. Feed Substrates

Mealworms were reared on wheat bran, rapeseed cake, brewers’ spent grain, and deproteinized grass. The wheat bran, used as the reference feed, was bought from Danish Agro, Karise, Denmark. The rapeseed cake, the leftover from oil extraction, was provided by DLG Food Oil, Dronninglund, Denmark. The brewers’ spent grain, leftover from beer fermentation, was provided by the Danish brewery, Søgaards Bryghus, Aalborg, Denmark. The deproteinized grass, leftover materials after partial extraction of the grass proteins, was provided by Ausumgaard, Hjerm, Denmark. The brewers’ spent grain and the deproteinized grass were dried at 105 °C until constant weight before use. All substrates were blended for about a minute in a kitchen blender and allowed to absorb moisture from the air in an incubator, reaching an average substrate moisture content of 22%.

BSF larvae were reared on chicken feed, rapeseed cake, brewers’ spent grain, deproteinized grass, and biopulp. The chicken feed was used as a reference feed substrate, bought from Danish Agro. The rapeseed cake was provided by DLG Food Oil. The brewers’ spent grain was provided by the Danish brewery, Hancock, Skive, Denmark. The deproteinized grass was provided by Ausumgaard. The biopulp, an aqueous slurry of catering waste, was provided by the waste handling company, Gemidan, Nørresundby, Denmark. The substrate moisture content in all cases was adjusted to 77% through the addition of hot tap water.

The proximate composition of fat, carbohydrate, protein, dietary fibers, and ash in the feed substrates were quantified by ALS, Humlebæk, Denmark, which is an accredited laboratory for the nutritional and chemical analyses of foods, beverages, and environmental samples. In brief, lipids were measured by NMR, carbohydrates were measured as monosaccharides on an HPLC after enzymatic digestion, proteins were measured by the Dumas method, ash contents were measured gravimetrically after combustion in an ashing furnace, and dietary fibers were estimated by subtraction.

### 2.3. Rearing Experiments

Mealworm rearing experiments were carried out in cylindrical containers (diameter = 7.5 cm, height = 11.5 cm) placed in a dark climate chamber at 28 °C, 85–90% humidity. A total of 65 individuals were added to each container along with 65 g (in terms of DW) of feed substrate. All rearing experiments were carried out in replicates of five. One additional replica per feed substrate was used as a sacrificial replica from which larvae (5–10 individuals at a time) were regularly sampled for DW determination.

BSF larvae rearing experiments were carried out in cylindrical containers (diameter = 7.5 cm, height = 11.5 cm) placed in a dark climate chamber at 28 °C. An open water reservoir inside the climate chamber maintained 90–95% humidity. A total of 70 larvae were added to each container along with 53 g of wet feed substrate (12 g of dried substrate). Another 43 g of wet feed substrate (10 g dried substrate) was added on Day 6 of the rearing experiment. All rearing experiments were carried out in replicates of five. Three additional replicas per feed substrate were used as sacrificial replicas. Larvae were sampled for DW determination from one replica at a time.

All rearing experiments were terminated when 5–10% of the mealworms had turned into pupae or 5–10% of the BSF larvae had turned into pre-pupae. The larvae were separated from the remaining mixture of feed substrate and frass, counted, euthanized at −18 °C, and stored frozen until further analysis.

### 2.4. Larval CO_2_ Production Rates

Larval CO_2_ production rates, $r_{CO2}$ were measured at 2–4-day intervals on 5–10 randomly selected larvae from each container. Measurements were carried out as quickly and gently as possible. Immediately after the larvae were collected, they were transferred to a 50 mL plastic centrifugation tube placed in a water bath at 28 °C. A pre-calibrated PS2110 Carbon Dioxide Gas Sensor (Pasco, Roseville, CA, USA) was inserted into the top of the tube, thus forming a closed respiration chamber. The CO_2_ concentration in the respiration chamber was recorded at a frequency of 1 Hz. The CO_2_ sensor was calibrated in a 1.18 L closed, cylindrical Perspex chamber with a magnetic stirrer bar at the bottom. The calibration chamber was filled with N_2_. Pulses of 0.2 mL CO_2_ were gradually injected after the signal on the sensor had stabilized. The sensor signal was linear up to at least 10,000 ppm CO_2_. The linear increase in CO_2_ concentration recorded for a period of 2 min, starting 1 min after the tube had been closed, was used as a measure of $r_{CO2}$. The larvae were weighed and returned alive to the same container from where they had been collected.

### 2.5. Analytical Procedures

The WW and DW of larvae were determined by weighing before and after drying at 60 °C in an oven until constant weight. Larval ash content was determined by weighing the dried larvae after incineration in a furnace at 550 °C and then compared to the larval DW. Before the rearing experiments were started, the total larval WW was determined, and when the experiments were terminated, the total larval WW, DW, and ash content. Larvae from the sacrificial replicas (17–19 evenly spaced sampling points for mealworms, and 5 for BSF larvae) were weighed (WW, DW, and ash content), 3–20 individuals at a time. The DW of larvae used for the CO_2_ measurements was estimated based on the DW contents of larvae from the sacrificial replicas on the same or adjacent sampling points.

The weights of the feed substrates were measured when the rearing experiments were started, and the leftover mixtures of feed substrates and frass when the experiments were terminated. Samples of the feed substrates and all the leftover feed substrates and frass mixtures were dried at 105 °C to quantify the feed substrate dry matter and substrate moisture contents.

Larval carbon and nitrogen contents were measured by loading 2–2.5 mg of dried ground larvae into a Thermo Fisher Flash Smart Elemental Analyzer (Thermo Fisher Scientific, Waltham, MA, USA). Measurements were compared to standards of 1, 2.5, and 3.5 mg of acetanilide. Larval protein contents were subsequently estimated based on the measured nitrogen contents using a conversion factor, *K_p_* = 4.67 [26].

The lipids from the larvae were extracted in accordance with the Bligh and Dyer protocol [27]. Between 0.05 and 0.2 g of dried and ground larvae was added to a glass tube along with enough water to reach 80% water content (W/W). Next, 2 mL of methanol and 1 mL of chloroform were added, and the mixture was shaken for 20 min. An additional 1 mL of chloroform was then added, followed by one more minute of shaking. To finally achieve a biphasic system, 1 mL of water was added. The bottom layer containing lipids and chloroform was decanted and left to evaporate overnight. The remaining lipids were then weighed, and the mass used to calculate the larval lipid contents.

### 2.6. Substrate Conversion Efficiency

Substrate conversion efficiencies, SCEs, were evaluated from a mass balance approach, comparing the overall gain of larval DW to the overall decrease in DW of the feed substrates,(1)$$SCE=\frac{n_{t}X_{DW,t}-n_{0}X_{DW,0}}{W_{DW,0}-W_{DW,t}}$$
where *n*_0_ and *n_t_* are the number of larvae at the start and at time of harvest, *X_DW_*_,*t*_ and *X_DW_*_,0_ are the larval dry weights at harvest and at the start of the rearing experiments, and *W_DW_*_,0_ and *W_DW_*_,*t*_ are the total dry matter of the substrates and the substrate/frass mixtures at the start and at harvest, respectively.

### 2.7. Growth and Metabolic Performance of Mealworms and BSF Larvae

Larval growth was modeled by the Verhulst logistic model as described in [24],(2)$$X=\frac{X_{max}}{1+\left(\frac{X_{max}-X_{0}}{X_{0}}\right)e^{-\mu_{max}\left(t-t_{0}\right)}}$$
where *X_max_* is the maximal weight of the larvae [mg], *X*_0_ is the weight of the starter larvae [mg] at *t* = *t*_0_ (larval age at onset of rearing experiment), and *μ_max_* is the maximal specific growth rate of the larvae [day^−1^], theoretically attained only in larvae approaching zero weight. Equation (2) was fitted to the measured DW of the BSF larvae to find the combination of *X_max_* (measured as DW) and *μ_max_* that resulted in the smallest difference (average mean error) between the model and measured values. It has been shown that the sigmoidal growth curves of BSF larvae are well-described by Equation (2) in a variety of feed substrates [5,24,25]. In mealworms, the growth curves are also sigmoidal [28] and have been described by logistic models [29]. The specific growth rate, *µ* [day^−1^], of different sized larvae was then estimated from Equation (3),(3)$$\mu =\mu_{max}\left(1-\frac{X}{X_{max}}\right)$$
and the growth rate of the larvae, *r_X_* [mg day^−1^], was described by(4)$$r_{X}=\mu X$$

In fast growing animals, respiration rates depend on growth as well as on maintenance [30]. Therefore, the rate of individual larval CO_2_ production, $r_{CO2}$ [mg CO_2_ day^−1^], should also reflect these two overall metabolic processes and was modeled by Equation (5).(5)$$r_{CO2}=Y\mu X+mX^{b}$$

The first expression on the right-hand side describes growth associated respiration as being proportional to the growth rate of the larvae, with *Y* [unitless] being the cost of growth. The second expression describes maintenance dependent CO_2_ production, where *m* is a maintenance coefficient, in units of [day^−1^], and *b* is a metabolic scaling coefficient. The specific CO_2_ production rate was thus estimated from Equation (6):(6)$$q_{CO2}=\frac{r_{CO2}}{X}=Y\mu +mX^{b-1}$$

In mealworms, $q_{CO2}$ was fitted to the specific larval respiration rates determined experimentally by finding the optimal combination of *Y*, *m*, and *b* that resulted in the smallest difference (average mean error) between the model and measured values. In BSF larvae, it has been shown that *b* = 1 [5,24,25], meaning that in this species, the maintenance rate is proportional to the dry weight of the larvae.

Feed assimilation rates, *r_A_* [mg day^−1^], which was not measured directly because of the complex matrix in which the larvae were reared, were estimated as(7)$$r_{A}=r_{X}+r_{CO2}$$
and the specific feed assimilation *a* was calculated from Equation (8)(8)$$a=\frac{r_{A}}{X}=\mu +q_{CO2}$$

Finally, for the carbon net growth efficiency, NGE* was calculated from Equation (9):(9)$${NGE}^{*}=\frac{\mu }{a}=\frac{r_{X}}{r_{A}}$$

Net growth efficiencies are mostly quantified in terms of energy, but here the star indicates that NGE* measures the efficiency of carbon transfer from the feed to larvae [22]. The average carbon net growth efficiencies of the larvae, NGE*_avg_, were estimated from Equation (10):(10)$${NGE}_{avg}^{*}=\frac{\int r_{X}}{\int r_{A}}=\frac{X_{t}-X_{0}}{\int r_{A}}$$
where *X_t_* is the weight of the larvae at the time of harvest.

To compare the modeled and measured rates of growth, CO_2_ production, and feed assimilation, all variables were expressed in terms of carbon equivalents when the model in Equations (2)–(8) was executed. Firstly, Equation (2) was fitted to the larval DW in order to estimate *X_max_* and *μ_max_*, and secondly, Equation (6) was fitted to specific larval respiration rates to estimate *Y*, *m*, and also *b* for mealworms. Goodness-of-fits were evaluated from the absolute mean errors between the modeled and measured values. Carbon contents were estimated based on the measured mass fractions of carbon in the larvae and the mass fraction of carbon in CO_2_ (0.29) [24]. To better compare the net growth efficiency to SCE, NGE*_avg_-values were translated into net growth efficiencies based on dry weight changes of the larvae and feed substrates, NGE*_DW,avg_, taking into account the mass fractions of carbon in the larvae and feed substrates. We assumed that the carbon fraction of the organic part of all feed substrates was 0.49, corresponding to the average carbon content of microbial biomass [31], and like what is also normally found in plant-based materials [32]. The measured variables and modeled parameters were compared by one-way ANOVA at the 5% probability level.

## 3. Results

### 3.1. Feed Substrates

The composition of fat, carbohydrate, protein, dietary fibers, and inorganic components (ash) in the feed substrates is shown in Table 1. Compared to the two base line substrates, wheat bran or chicken feed, all were rich in protein, all but deproteinized grass were rich in fat, and all but biopulp were low in carbohydrates. All substrates also contained variable fractions of dietary fibers (the deproteinized grass consisted largely of fragmented grass leaves), presumably of little nutritional value to the insect larvae.

### 3.2. Growth and Metabolic Performance of Mealworms

The growth curves and CO_2_ production from mealworms reared on wheat bran, rapeseed cake, brewers’ spent grain, and deproteinized grass are shown in Figure 1. Biopulp was not offered to mealworms due to its high moisture content. The mealworms grew fastest and became largest on wheat bran, followed by rapeseed cake (Table 2). The experiments were terminated when the mealworms began turning into pupae at 65 or 93 days of age. The mortality was low on both feed substrates. Growth was slow on brewers’ spent grain and no larvae had reached the pupal stage at 87 days of age when this experiment was terminated. These mealworms also had low mortality. The mealworms reared on deproteinized grass grew even slower, and their mortality reached 100% after 42–66 days in four out of five replicate cultures. In the fifth culture, the mortality was 80% on Day 90 when this experiment was terminated, and no larvae reached the pupal stage. Growth was close to exponential until the larvae reached 15–30 mg DW. From the linear regression analysis of the linear parts of plots of *ln*-transformed larval DW vs. time (Appendix A), the specific growth rates were determined as 0.10 day^−1^ (Days 14–49), 0.06 day^−1^ (Days 14–56), 0.02 (Days 14–75), and 0.01 day^−1^ (Days 14–93) on wheat bran, rapeseed cake, brewers’ spent grain, and deproteinized grass, respectively (Table 2).

The logistic model was able to describe the increase in mealworm weight on all feed substrates (Figure 1, left panels). Maximal specific growth rates and maximal DWs could therefore be estimated by fitting Equation (2) to the experimental data (the data did not allow for the estimation of the maximal DW on deproteinized grass because the larvae were still growing exponentially when harvested, Appendix A), and the specific growth rates across the experiments, calculated from Equation (3) (Table 2). The maximal specific growth rate is, in principle, attainable only by larvae approaching zero weight, and therefore a bit higher than the measured specific growth rates. The maximal DW of the larvae (7–78 mg, Table 2) represents the DW that the larvae could have theoretically attained if they had not been harvested when the first pupae formed and is thus higher than the measured DWs.

The CO_2_ production rate of the mealworms increased with time, to level off and even decrease in wheat bran 1–2 weeks before the first pupae were observed (Figure 1, central panels). CO_2_ production was substrate dependent, increasing fastest with weight in mealworms grown on wheat bran (Figure 2A). Specific respiration rates were highest (0.2–0.3 day^−1^ on all substrates) when the mealworms were the smallest (Figure 2B). The smallest mealworms with DWs around 1 mg thus released a daily amount of carbon in the form of CO_2_, corresponding to 20–30% of their own carbon content. As the mealworms became larger, their specific respiration rates decreased below 0.05 day^−1^. These changes in specific respiration rates could be described by Equation (6) on all feed substrates, and the estimated costs of growth, maintenance coefficients, and allometric coefficients, *b*, are shown in Table 2. For each mole of carbon incorporated into the mealworms, 0.36–1.08 moles of carbon were released as CO_2_ as the cost of growth. Maintenance coefficients ranged from 0.02 to 0.05 day^−1^. Thus, a mealworm containing 1 mmol carbon (approximately 20 mg DW since carbon (12 mg) makes up 59–60% of the larval DW, Table 2) daily metabolized an amount of carbon corresponding to 2–5% of its own carbon content to sustain non-growth-related metabolic processes, which was more in smaller individuals and less in larger ones. Estimates of *b* ranged from 0.43 to 0.53, indicating a strong relationship between maintenance and DW. This is demonstrated as nonlinear relationships between specific CO_2_ production rates and specific growth rates (Figure 1, right panels). The specific feed assimilation rate, estimated from Equation (8) (Figure 1, central panels), reached values of up to 0.34, 0.37, 0.17, and 0.16 day^−1^ in the smallest individuals 19 days of age, on wheat bran, rapeseed cake, brewers’ spent grain, and deproteinized grass, respectively. The specific feed assimilation rates decreased as the larvae grew older, close to levels equivalating the maintenance metabolism.

Because the specific rates of feed assimilation, growth, and CO_2_ production are weight dependent, so is the NGE* (Figure 2C). Highest values (0.4–0.5) were observed in mealworms (20–60 mg DW) reared on wheat bran and rapeseed cake (10–40 mg DW). Smaller mealworms had a lower NGE* because of their high maintenance metabolism, while larger ones had a lower NGE* due to their low specific growth rates. Most biomass production in the mealworms took place in periods with high NGE*. At the time of harvest, the NGE*_avg_ attained values of 0.40, 0.37, 0.17, and 0.16 on wheat bran, rapeseed cake, brewers’ spent grain, and deproteinized grass, respectively (Table 2). The differences between the NGE*_avg_ and NGE*_DW,avg_ were caused by the uneven carbon contents of the larvae and feed substrates. It was in the feed substrates with high NGE*_avg_ that we also observed the highest SCEs (0.14 and 0.12). Zero SCE on deproteinized gras was a result of the high mortality. There was no net gain of larval mass despite the surviving individuals increasing in weight. Start and end masses and carbon balances on all feed substrates are shown in Appendix A.

The more than 10 times difference in the final DW at the time of harvest did not seem to result in systematic differences in the macromolecular body composition of the mealworms (Table 2). Proteins made up close to half of the DW or above, and were highest in the smallest ones. Lipids and inorganic components made up less than 30% and 2% of the DW of mealworms reared on wheat bran or rapeseed cake, respectively, and were not quantified on the remaining substrates because too little mealworm biomass was produced to allow for all components to be analyzed. Carbon made up 59–60% of the larval DW on wheat bran, rapeseed cake, and brewers’ spent grain, and a bit less (53%) on deproteinized grass.

### 3.3. Growth and Metabolic Performance of BSF Larvae

Growth of the BSF larvae is shown in Figure 3. Growth was fastest and the larvae became largest on chicken feed, rapeseed cake, and biopulp (Table 3). Brewers’ spent grain and deproteinized grass resulted in slower growth and smaller larvae. The mortality was low, and the larvae began forming prepupae on all substrates at 23–25 days of age. Growth followed the logistic model (Equation (1)), and the specific growth rates thus decreased as the larvae grew (Figure 3, left panels). The actual specific growth rates were determined from Appendix A. On most of the feed substrates, the CO_2_ production rates increased substantially with time and stabilized when the BSF larvae reached 15–17 days of age. Specific CO_2_ production rates were highest in the smallest larvae, decreased with time, and could be described by Equation (6) keeping the coefficient *b* = 1. Only on brewers’ spent grain did the specific CO_2_ production rate stay about the same throughout the experiment. Specific feed assimilation rates were, on all feed substrates, highest in the smallest larvae. Maximal DWs and specific growth rates, predicted from Equation (1) (Figure 3, left panels) are shown in Table 3. Because the coefficient *b* = 1, the maximal feed assimilation rate, *a_max_*, could theoretically be predicted by substituting the maximal specific growth rate into Equations (6) and (8) [24]. Close to linear relationships between specific CO_2_ production and specific growth rates (Figure 3, right panels) were consistent with *b* = 1 in the BSF larvae. This means that the maintenance rate of the BSF larvae remains proportional to their weight across their life span.

Like in the mealworms, the CO_2_ production rates increased with the weight of the BSF larvae but stabilized before they reached the maximal DW (Figure 4A). The specific CO_2_ production rates decreased continuously with larval DW (Figure 4B). NGE* was highest in the smallest larvae and decreased with weight (Figure 4C). NGE* was considerably higher in the larvae reared on chicken feed, rapeseed cake, and biopulp than in those reared on brewers’ spent grain or deproteinized grass. The same was the case for the NGE*_avg_. The highest SCEs were seen on the same substrates that resulted in a high NGE*_avg_. The start and end masses and carbon balances on all feed substrates used to evaluate the SCEs are shown in Appendix A. On all substrates, as was also the case for mealworms, the SCE was smaller or equal to the NGE*_DW,avg_ (Figure 5).

The macromolecular body composition of the BSF larvae at the time of harvest is seen in Table 3. Additional measurements taken while the larvae were growing are shown in Appendix A. As was the case for the mealworms, the differences in weight of the BSF larvae did not result in systematic differences in their body composition, except for their dry weight contents, which was lowest in the smallest ones reared on brewers’ spent grain or deproteinized grass. Dry weight contents also increased with age. Proteins were also the dominating constituent in this species, followed by lipids (the lipid content even surpassed the protein content of the BSF larvae reared on biopulp). The larvae reared on deproteinized grass were particularly low in lipid and rich in inorganic materials. Carbon made up 54–55% of the larval DW on four out of five substrates, but only 44% on deproteinized grass, likely because these larvae were particularly rich in inorganic components and low in lipids.

## 4. Discussion

The mealworms and BSF larvae grew, at least to some extent, on all the feed substrates they were offered. Both species were reared on a selection of feed substrates to create variation in their growth patterns. Apart from the chicken feed, these were either by-products from established (wheat bran, rapeseed cake, and brewers’ spent grain) or upcoming (deproteinized grass) agricultural or food products or recycled catering waste (biopulp). All are also potential feed resources for insects at large scale, representing considerable variety in composition and nutritional quality (Table 1). The specific growth rate and maximal weight of the larvae of both species were highly substrate dependent, as were their feed efficiency in terms of SCE and NGE*_avg_ (Table 2 and Table 3). The mealworms grew best on wheat bran followed by rapeseed cake and seemed to be the most fastidious of the two species. There was almost a six times difference between the maximal specific growth rates and eleven times difference between the maximal DW on the four feed substrates, and only a few mealworms remained alive on deproteinized grass (Table 2). The BSF larvae grew on all five feed substrates, with just a three times difference between the highest and lowest maximal specific growth rates and maximal DW (Table 3). Equations (2) and (6) were able to describe the larval DWs and specific CO_2_ production rates on all feed substrates, allowing us to characterize and compare the performances of fast as well as slow growing larvae of both species.

Both species were presumably reared under close-to-optimal conditions, except for variations in feed substrate quality. The mealworms reached DWs of 63 and 44 mg in 65 or 93 days on wheat bran or rapeseed cake, respectively (Figure 1, left panels), and might presumably have reached 79 and 56 mg, respectively (Table 2). These values are well in line with former observations in mealworms reared on wheat bran and other agricultural products [14,15,16,17]. For the BSF larvae, the measured and predicted DWs up to around 120 mg and maximal specific growth of 0.5–0.77 day^−1^ are also in line with former studies [5,24,25]. To minimize the risks of feed shortage, at least for the reference feed substrates, we offered larger feed quantities (1 g per mealworm and 0.3 g per BSF larva in terms of dry matter) than they would expectedly utilize. These values are above the recent recommendations of 0.21 g feed per mealworm [33] and 0.22 g feed per BSF larva [34] for use in experimental studies, and our larvae also achieved considerably higher weights than those reared according to the recommendations (up to 185 mg vs. 83–117 mg WW for mealworms and 116 mg vs. 17–51 mg DW for the BSF larvae). We also did not fully adhere to recent recommendations of using water-saturated substrates due to variability in their water-holding capacities [35]. For convenience, we used the same initial substrate moisture content in all feeding substrates given to the BSF larvae, as their metabolic performance is not appreciably affected by the water saturation level [5]. Furthermore, the substrate moisture content was so high that it presumably should have minimized the aerobic microbial substrate degradation during the experiments.

The specific CO_2_ production rates of the BSF larvae, roughly 0.1–0.5 day^−1^ (Figure 3, center panels and Figure 4B) fell in the same order of magnitude as previously reported [5,24,25]. The same was the case for the mealworms (Figure 1, center panels and Figure 2B). One former study found CO_2_ production rates in mealworms between 9 and 23 μL g^−1^ day^−1^, depending on growth [36]. This corresponds to specific CO_2_ production rates of 0.07–0.17 day^−1^, assuming a 40% DW content containing 59% carbon. A second study compared the O_2_ uptake rates of fed and starved mealworms [37], which at 30 °C were 500 and 1200 μL g^−1^ h^−1^ for starved and fed individuals weighing 75 mg, respectively. This corresponds to specific CO_2_ production rates of 0.06–0.15 day^−1^, assuming a respiratory quotient of 0.75 and 35% DW content (quantified in that study) and 59% carbon content. Compared to these studies, we observed larger differences between the highest and lowest specific CO_2_ production rates (roughly 0.03–0.3 day^−1^), reflecting that our measurements were taken on mealworms spanning almost their full weight range.

In terms of feed efficiency, the BSF larvae generally performed better than the mealworms. The same was concluded in a second study comparing different performance indicators of these two species reared on food waste [38]. The NGE*_avg_ ranged between 0.16 and 0.40 in the mealworms (Table 2) and 0.36–0.56 in the BSF larvae (Table 3). The feed substrates that resulted in high NGE*_avg_ were also those with the highest SCE (Table 2 and Table 3). Thus, larval performance was a primary determinant for the SCE, at least on most of the feed substrates. Formally, NGE*_avg_ and SCE are directly comparable only if the larvae and feed substrates have the same elemental composition, since the former describes the efficiency of retaining carbon, and the latter that of the DW. Therefore, we also estimated the average net growth efficiency based on the dry matter changes of larvae and feed substrates, NGE*_DW,avg_, which is directly comparable to the SCE (Figure 5). In some feed substrates, the SCE was considerably smaller than the NGE*_DW,avg_. This shows that not only the performance of the larvae influences the outcome. The most likely explanation is that competing microbial activities contributed to the degradation of the feed substrates [5]. This seems to be common, at least in BSF larval cultures [22]. The largest differences between NGE*_DW,avg_ and SCE were seen in mealworms (Table 2 and Figure 5), which is likely due to the long rearing periods and comparatively dry feed substrates that oxygen can most easily penetrate [5]. The amount of feed substrate relative to the number of larvae also probably affected the SCE, since surplus feed components remain available for microbial degradation. Visible surface layers of mold developed during the experiments, potentially also affecting the metabolism of the mealworms. Thus, Figure 5 also illustrates that the feed efficiency of insect larvae (NGE*_avg_) can in some cases differ considerably from the feed efficiency of their overall rearing processes (SCE).

The carbon balances in Appendix A confirm that the SCE and NGE*_DW,avg_ determinations are generally reliable. In feed substrates where the SCE is comparable to NGE*_DW,avg_, the removal of carbon from the feed substrate should correspond to the sum of carbon stored in the larvae and the CO_2_ they have produced. This was also the case for BSF larvae grown on chicken feed and biopulp (Figure 5), where we recovered 110% and 104% of the removed carbon, respectively (Appendix A). However, the fact that we recovered more carbon than consumed also shows that our data were subjected to some degree of uncertainty, which may stem from uncertainties in, among other things, the carbon contents of the feed substrates and larvae, that we do not know which feed components the larvae metabolize, the accumulation of carbonates and volatile fermentation products in feed substrates [5], and uncertainties in the CO_2_ determinations including possible stress effects when the larvae are placed in a respiration chamber [5,24]. In feed substrates where the SCE is lower than the NGE*_DW,avg_, the sum of carbon in the larvae and their accumulated CO_2_ production should be less than what is removed from the feed substrate. This was indeed the case in both the mealworms (Appendix A) and BSF larvae (Appendix A). The greater the difference between the SCE and NGE*_DW,avg_ (Figure 5), the less carbon from the feed substrate is recovered.

The BSF larvae had metabolic advantages over the mealworms. Higher specific growth rates made growth-related metabolic processes more dominant in this species compared with the mealworms. Therefore, the NGE* reached higher values in the BSF larvae (0.7–0.8 on the best quality substrates, Figure 4C) than in the mealworms (0.4–0.5 on the best quality substrates, Figure 2C). Fast growth also means that the BSF larvae reach their maximal weight in a short period of time. Thus, the BSF larvae must invest in maintenance for just a short period of time. Former studies reported NGE*_avg_ values as high as 0.62–0.63 in BSF larvae [5,24], which were not much different from the highest values found here (Table 3). The mealworms had advantages over the BSF larvae in terms of the allometric response of maintenance to larval weight. In both species, the NGE* decreases when the larvae approach their maximal weight because their specific growth rate also then decreases, but less rapidly in the mealworms because they, at the same time, reduce their maintenance rate. Although the smallest mealworms had the highest maintenance rates of all and thus also low NGE* values (Figure 2C), the effect on NGE*_avg_ was low. Only a small percentage of the weight was gained before the NGE* increased. However, one challenge associated with the high maintenance rates of small mealworms may be that they, in this part of their life, are particularly dependent on high specific feed assimilation rates to create the nutritional surplus that allows them to grow.

The experiments allowed us to estimate the costs of growth and the maintenance coefficients, although the accuracy faced different limitations in the two species. To our knowledge, this is the first time that the costs of growth and maintenance coefficients were quantified in larvae of the yellow mealworm beetle across their life span. Growth and CO_2_ production were therefore followed tightly in this species. However, the cost of growth, *Y*, the maintenance coefficient, *m*, and the allometric coefficient, *b*, had to be estimated simultaneously by fitting Equation (6) to the measured specific CO_2_ production rates using nonlinear regression analysis (Figure 1, center panels). Several combinations of *Y*, *m*, and *b* fit the model quite well to the data. Thus, we cannot rule out that the actual values may have differed somewhat from our best estimates (Table 2). In the BSF larvae, Equation (6) was fitted to the data, keeping *b* = 1, making the estimates of *Y* and *m* more robust. In mealworms, the cost of growth ranged from 0.36 to 1.08, highly similar to former estimates in BSF larvae, from 0.34 to 0.44 on chicken feed or brewery waste, and higher on poorer substrates [5,24,25]. In the BSF larvae, the cost of growth ranged from 0.12 to 0.73 (Table 2 and Table 3). This was, in most cases, lower than seen before, and likely underestimates the actual values. The BSF experiments were not designed optimally to measure this parameter. Since the metabolic performance of BSF larvae has been quantified several times before [5,24,25], measurements of the larval CO_2_ production rates were only initiated after the BSF larvae reached 11 days of age, in order not to unnecessarily stress the smallest and most delicate BSF larvae in the respiration chamber. Thus, the BSF larvae experiments had lower numbers of coherent measurements of larval weights and CO_2_ production rates compared with the mealworm experiments, and none in the first phases of their life when the specific growth rate is the highest, and the cost of growth most dominant. Differences also arose from former values being calculated based on a 49% carbon content in the organic part of the larvae, corresponding to the average of microbial biomass [31]. Many biological materials have similar carbon contents, and thus many feed substrates, but higher values (0.57–0.61) are found in materials of animal origin [32]. We also found higher carbon contents in both the BSF larvae and mealworms (Table 2 and Table 3), close to the level (0.56–0.66) previously seen in BSF larvae [39] as well as in oleaginous microorganisms [40]. The cost of growth of the mealworms reared on wheat bran and the BSF larvae reared on deproteinized grass seemed surprisingly high, and was suspiciously low in the BSF larvae reared on rapeseed cake. Still, it seems that the cost of growth of larvae of both species in the most suitable substrates amounted to 0.27–0.39. One study [41] measured the costs of growth in a cockroach and caterpillar to 6.9 and 0.34 kJ g^−1^ DW, respectively, and compiled an overview of the costs of growth of fish, amphibians, reptiles, birds, and mammals ranging from 5 to 22 kJ g^−1^ DW. In terms of carbon equivalents (the combustion energy of dry biomass is 22.7 KJ g^−1^ [31]), these costs of growth correspond to 0.015 (surprisingly low value) and 0.3 for the two insects, and 0.22–0.97 for the rest, with an average of 0.4. Thus, the costs of growth of mealworms and BSF larvae appear to be like those of other animals, potentially in the lower range, and are probably more influenced by the quality of the feed substrate than by species-specific differences.

Maintenance depended on the quality of the feed substrate, the larval species (weight specific maintenance rates were generally the highest in the BSF larvae despite their larger size), and was differently affected by the weight of the larvae. Maintenance dependent differences in feed efficiency may thus relate to both the substrate and the species of the insect. In mealworms, the maintenance coefficient ranged from 0.02 to 0.05 day^−1^ (Table 2) but was weight dependent (*b* < 1). Allometric relationships have been described in many areas of physiology and metabolism, with *b* most often close to 0.75 [42]. The lower *b* values (0.43–0.53) indicate particularly strong, inverse relationships between the maintenance rate and weight in mealworms. Equation (6) predicted that the starter mealworms (DW = 1 mg and C content = 0.04 mmol, specific growth rate, *μ* = 0.11 day^−1^) had a maintenance rate of 0.11 day^−1^ on wheat bran, which dropped to just 0.02 day^−1^ at the time of harvest (DW = 65 mg and C content = 2.6 mmol, *μ* = 0.02 day^−1^). On rapeseed cake, the predicted maintenance rate similarly dropped from 0.27 day^−1^ in the starter mealworms to 0.03 day^−1^ at the time of harvest. It is, however, not the first time such low allometric coefficients have been described in mealworms. One former study [43] found that allometric coefficients associated with water conductance and water adsorption capacity were even below 0.4. Quite the opposite is seen in BSF larvae. In this species, maintenance coefficients seem weight independent (*b* = 1). The maintenance coefficients on four of the substrates (0.07–0.13 day^−1^) aligned with former estimates on chicken feed and brewery waste (0.05–0.13 day^−1^) [5,24,25], despite the former values also being based on 49% carbon in the organic fraction of the larval biomass. Only the BSF larvae reared on brewers’ spent grain had a considerably higher maintenance coefficient (0.21 day^−1^), suggesting that this feed substrate may have stressed the larvae [24].

The maximal specific feed assimilation rates, based on carbon equivalents, were close to 1 day^−1^ in the BSF larvae on all feed substrates (Table 3). This agrees well with earlier estimates in BSF larvae [5,24,25], taking into account that these former studies based their estimates on a lower carbon content than measured here. The differences in growth patterns therefore did not stem from differences in their ability to assimilate feed components from the different feed substrates. In contrast, the feed assimilation rates in the mealworms were quite different on some of the feed substrates (Table 2), and at least 2–3 times lower than in the BSF larvae. The gap between the curves describing the specific feed assimilation and respiration rates in Figure 1 and Figure 3, center panels, represent the fraction of assimilated feed, measured in carbon equivalents, that at each time point was guided into growth. This fraction remained fairly stable in mealworms at all weights. Although the specific feed assimilation rate was the highest in the smallest individuals, so was the maintenance. Small BSF larvae were, in contrast, able to accumulate larger fractions of their assimilated feed for growth than larger ones because of their high feed assimilation rates. When they grew bigger, the specific feed assimilation rate dropped, and several days before the first prepupae appeared, almost all of the feed assimilated seemed to be consumed by maintenance, which remained considerably higher than it did in the largest mealworms. Thus, the differences in specific growth rates between the BSF larvae and mealworms can be attributed to a combination of higher specific feed assimilation rates and differences in maintenance metabolism. If only the BSF larvae are considered, high specific feed assimilation rates were linked to high specific growth rates on chicken feed and rapeseed cake, but not on deproteinized grass. This substrate seemed to cause extraordinarily high costs of growth, seemingly forcing the BSF larvae to metabolize particularly large fractions of the assimilated feed to cover energy demands, leaving less for growth. Maximal specific feed assimilation and thus the growth rates were comparatively low on brewers’ spent grain and biopulp. Brewers’ spent grain also resulted in an extraordinary high maintenance coefficient, leading to particularly low specific growth rates on this feed substrate.

The macromolecular body composition of the larvae was to some extent influenced by the feed substrate. This is not unexpected for BSF larvae [44], while mealworms may regulate their body composition more tightly [45]. In mealworms, the lipid and protein contents remained within ranges also observed in other studies [26,46] and did not seem to differ much on the two best substrates, wheat bran or rapeseed cake (Table 2). The body composition of the BSF larvae varied considerably on different feed substrates (Table 3) but the protein and lipid contents remained within known ranges [47]. The protein content of the BSF larvae showed less variation than that of the lipid content but was determined with some degree of uncertainty. Chitin and other components, in addition to the proteins, contribute to the nitrogen content of the larvae, and different *K_p_* values have been recommended for insect larvae [26,48,49]. The BSF larvae were richest in inorganic components (ash), showing a great deal of variability. Since the body composition of the larvae were not always the same, we cannot exclude that the carbon content of the larvae changed during growth, affecting the estimated NGE*_avg_ values [5]. We were unsuccessful in producing enough starter mealworms to allow for the ongoing removal of individuals for carbon and nitrogen analyses, but measurements on growing BSF larvae suggest a rather limited age dependent variability (Appendix A). However, since most of our carbon measurements were in the range of 55–60% of larval DW, earlier studies may have underestimated the NGE*_avg_ in BSF larvae, since these studies assumed only 49% carbon in the organic part of the larvae [5,24,25].

Despite the feed substrates being varied in composition, giving rise to variable metabolic performances and body compositions of the larvae, this study shows, in line with former ones [20], that a variety of organic waste and by-products from major agricultural or food commodities can make up suitable feed substrates for BSF larvae, and also maybe, to a narrower extent, for mealworms. Both species grew fastest on the reference substrates wheat bran or chicken feed, but their feed efficiency, in terms of SCE as well as NGE*_avg_, reached comparable or even higher values on other feed substrates (Table 2 and Table 3). Further optimization could possibly be achieved by mixing organic resources [14,44]. Here, the organic waste and by-products were fed one at a time to provide variation in the substrate qualities to allow for the characterization of the metabolic performance of larvae showing a variety of growth patterns. Rapeseed cake alone supported the growth of both species. This has been previously demonstrated in BSF larvae [50], while mealworms have been reared successfully on mixed feed substrates containing rapeseed meal [51,52]. Biopulp is potentially a novel feed substrate, and the BSF larvae grew well on this waste-product. Other types of municipal organic waste are also well-accepted by BSF larvae [53]. The brewers’ spent grain and the deproteinized grass seemed to be unsuitable feed substrates for larvae of both species (the poor performance of BSF larvae on brewers’ spent grain was unexpected, therefore we used a different brewery as a supplier for the mealworm experiments). However, the quality of brewers’ spent grain for insect cultivation may vary a lot. Some studies have found that BSF perform well on brewers’ spent grain [54,55], while others noted the opposite [56,57], and some have observed that performances depend on the leftover type from the breweries [58,59]. Mealworms may also grow well on mixed feed substrates containing brewers’ spent grain [14,16,18]. Deproteinized grass has, to our knowledge, not been fed to mealworms or BSF larvae before, but BSF have been reared on milled grass clippings [3]. The quality of the deproteinized grass was sufficient to allow the BSF larvae to complete the larval stage and become prepupae, but its high content of recalcitrant straw materials probably prevented it from being utilized efficiently by either species. The differences in the feed substrate compositions apparently also affected the microbial activities, resulting in uneven differences between the NGE*_DW,avg_ and SCE (Figure 5). This, likely in interplay with larval activities, since SCE was affected the least, in the feed substrates that supported the highest larval weights and most rapid growth. Therefore, more attention to the microbial activities in the feed substrates may well hold further potential for optimization in insect farming.

## 5. Conclusions

The BSF larvae performed best on most by-products from the agricultural and food industries, showing lesser variation in maximal weight and maximal specific growth rate on different feed substrates than the mealworms. Additionally, the feed assimilation rate was less affected by the type of feed substrate in the BSF larvae compared with the mealworms. It was also the BSF larvae that, in general, converted the assimilated feed into their own biomass most efficiently, with an NGE*_avg_ of 0.33–0.56 across their life span. In mealworms, the NGE*_avg_ values were lower, but overlapped in the range of 0.16–0.40. Large differences between the NGE*_DW,avg_ and SCE in some feed substrates indicate microbial substrate degradation in parallel with larval activity. High similarities between the NGE*_DW,avg_ and SCE in other feed substrates, especially those supporting high larval weights and rapid growth, show that this is not always the case, at least if the moisture content of the substrates is high. The SCE may then approach its upper limit, determined by the NGE*_DW,avg_, and thus limited by the metabolic performance of the larvae. The feed substrates affected the metabolic rates of both species, and differences in the underlying metabolic processes gave both species advantages compared with each other, with respect to gaining the highest feed efficiency. The BSF larvae were advantageous in terms of the highest maximal specific growth and feed assimilation rates and the shortest development period, but disadvantageous in terms of maintenance. In contrast to the BSF larvae, the specific maintenance rate was weight dependent in the mealworms, and lowest in the largest individuals. The combined outcome of these differences in metabolic rates suggest that BSF larvae generally, but not universally, possess the highest potential to convert assimilated feed from most waste and by-products into new biomass, and thus the broadest potential to contribute to high feed efficiency in insect farming on different feed substrates.

## Figures and Tables

**Figure 1 animals-15-00233-f001:**
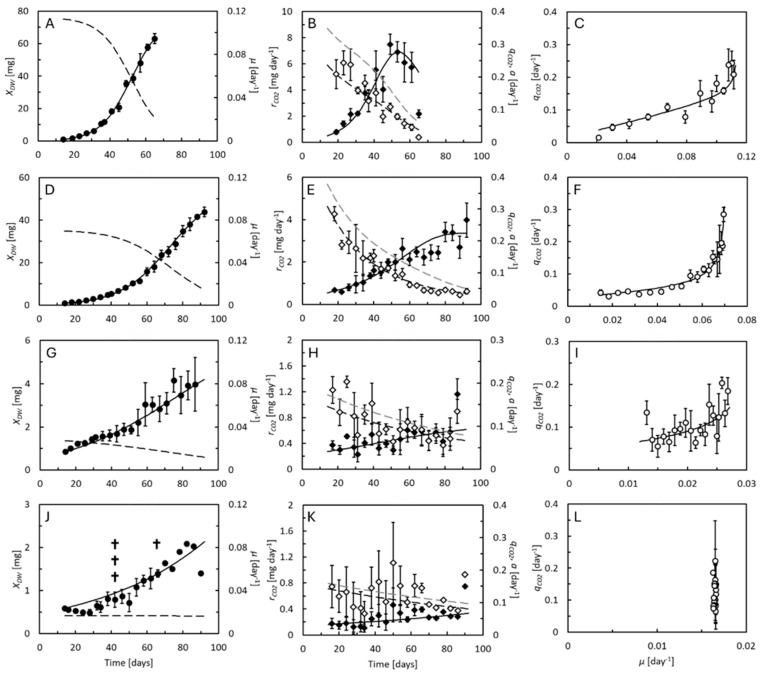
Growth and CO_2_ production rates of mealworms reared on (**A**–**C**) wheat bran, (**D**–**F**) rapeseed cake, (**G**–**I**) brewers’ spent grain, and (**J**–**L**) deproteinized grass. Left panels show the larval dry weight, *X_DW_* (●, solid curves predicted by Equation (2)), and specific growth rate, *μ* (dotted curves predicted by Equation (3)). Center panels show the CO_2_ production rates per individual larvae, $r_{CO2}$ (♦, solid curves predicted by Equation (5)), specific CO_2_ production rate (
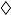
, dashed curves predicted by Equation (6)), and specific feed assimilation rate, *a* (dashed curves, grey scale predicted by Equation (8)). Right panels compare the specific CO_2_ production rate to specific growth rate (curve predicted by Equations (2)–(8)). Data points represent averages of 5 replicate cultures ± standard deviation. Extinction of individual larval cultures indicated by †.

**Figure 2 animals-15-00233-f002:**
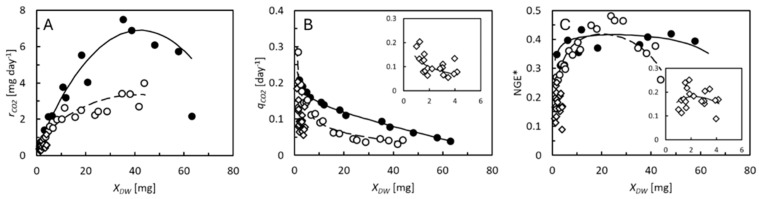
(**A**) Individual CO_2_ production rate, $r_{CO2}$. (**B**) Specific CO_2_ production rate (measured in carbon equivalents), $q_{CO2}$. (**C**) NGE* of mealworms reared on wheat bran (●, solid curve), rapeseed cake (○, dashed curve), or brewers’ spent grain (
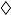
). Insets in panels (**B**,**C**) show the $q_{CO2}$ and NGE* on brewers’ spent grain on expanded scales. Data points represent average values of 5 replicate cultures (standard deviations indicated in Figure 1). Curves are model predictions from Figure 1.

**Figure 3 animals-15-00233-f003:**
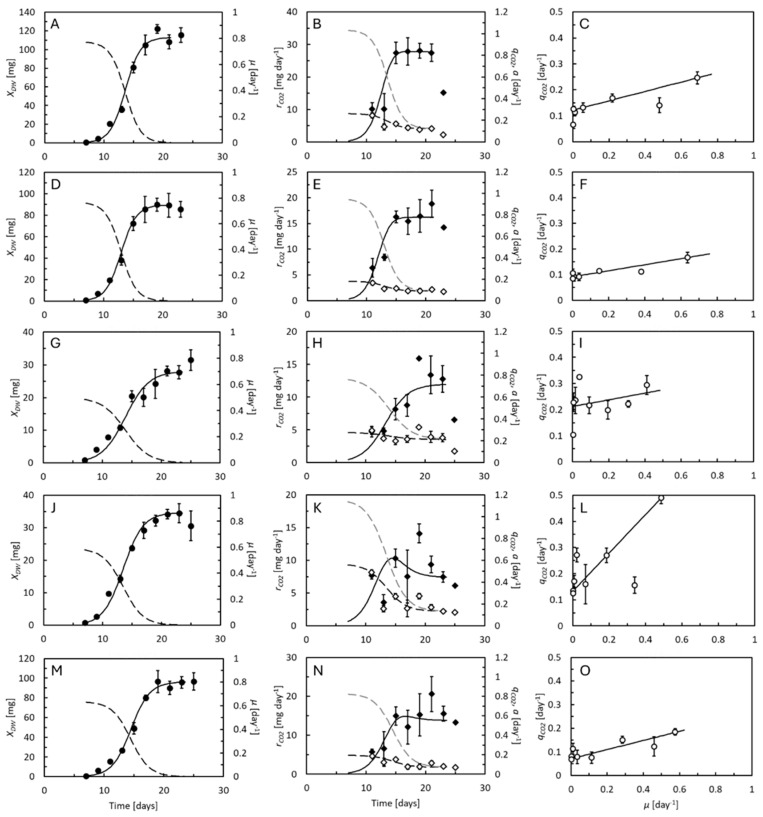
Growth and CO_2_ production rates of BSF larvae reared on (**A**–**C**) chicken feed, (**D**–**F**) rapeseed cake, (**G**–**I**) brewers’ spent grain, (**J**–**L**) deproteinized grass, and (**M**–**O**) biopulp. Left panels show the larval dry weight, *X_DW_* (●, solid curves predicted by Equation (2)), and specific growth rate, *μ* (dotted curves predicted by Equation (3)). Center panels show the CO_2_ production rates per individual larvae, $r_{CO2}$ (♦, solid curves predicted by Equation (5)), specific CO_2_ production rate (
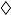
, dashed curves predicted by Equation (6)), and specific feed assimilation rate, *a* (dashed curves, grey scale predicted by Equation (8)). Right panels compare the specific CO_2_ production rate to specific growth rate (curve predicted by Equations (2)–(8)). Data points represent averages of 5 replicate cultures ± standard deviation.

**Figure 4 animals-15-00233-f004:**
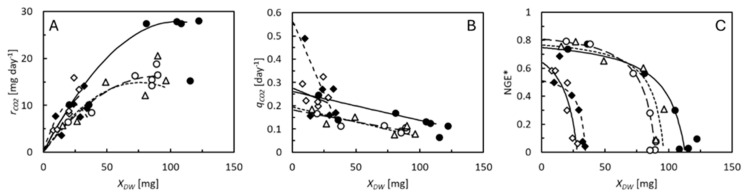
(**A**) Individual CO_2_ production rate, $r_{CO2}$. (**B**) Specific CO_2_ production rate (measured in carbon equivalents), $q_{CO2}$. (**C**) NGE* of BSF larvae reared on wheat bran (●, solid curve), rapeseed cake (○, dashed curve), brewers’ spent grain (
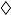
, solid curve), deproteinized grass (♦, dashed curve), and biopulp (Δ, dotted curve). Data points represent average values of 5 replicate cultures (standard deviation indicated in Figure 3). Curves are model predictions from Figure 3.

**Figure 5 animals-15-00233-f005:**
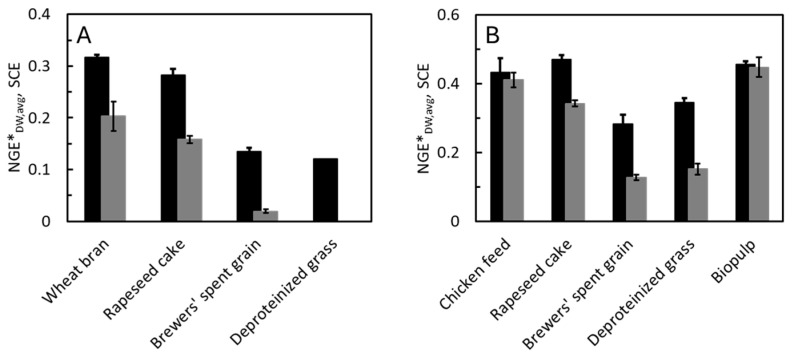
Net growth efficiency based on mass, NGE*_DW,avg_, black bars, and substrate conversion efficiency, SCE, grey bars of (**A**) mealworms and (**B**) BSF larvae reared on different feed substrates. Error bars indicate standard deviations of 5 replicate experiments.

**Table 1 animals-15-00233-t001:** Proximate composition of the feed substrates used for rearing the mealworms and/or BSF larvae.

Substrate		Wheat Bran ^1^	Chicken Feed ^2^	Rapeseed Cake ^1,2^	Brewers’ Spent Grain ^1^	Brewers’ Spent Grain ^2^	Deproteinized Grass ^1,2^	Biopulp ^2^
Component	Unit							
Fat	% DW	3.5	0.8	14.0	9.3	14.6	1.9	13.7
Carbohydrate	% DW	32.9	55.7	11.5	7.8	0.0	29.9	33.5
Protein	% DW	14.9	18.4	30.8	26.3	26.3	23.7	22.6
Dietary fibers	% DW	45.7	21.9	36.6	53.1	58.4	37.0	21.2
Ash	% DW	2.9	3.3	7.2	3.5	3.9	7.6	8.9

^1^ Feed substrates used for mealworms (brewers’ spent grain from Søgaards Bryghus). ^2^ Feed substrates used for BSF larvae (brewers’ spent grain from Hancock).

**Table 2 animals-15-00233-t002:** Experimental variables, model parameters, and tissue composition at harvest of mealworms reared on different feed substrates. Dry weight of larvae, age 14 days (first measurement), *X_DW_*_,14_. Time until >5% of the mealworms developed into pupae, *t_p_*. Specific growth rates, *μ* (determined from Appendix A). Feed assimilation rate at the start of the experiment, *a_Day_*_19_ (Equation (8)). Substrate conversion efficiency, SCE (Equation (1)). Maximal DW of larvae, *X_DW,max_*, and maximum specific growth rate, *μ_max_* (Equation (2)). Cost of growth, *Y*, maintenance coefficient, *m*, and allometric coefficient, *b* (Equation (6)). Average carbon net growth efficiency, NGE*_avg_ (Equation (10)), and the average net growth efficiency based on mass, NGE*_DW,avg_. Wet weight, *X_WW_*. DW fraction, *δ_DW_*. Contents of carbon, *δ_C_*, nitrogen, *δ_N_*, proteins, *δ_protein_*, lipids, *δ_lipid_*, and inorganic components (ash), *δ_ash_* in mealworms at time of harvest. Measurements represent averages of 5 replicate cultures ± standard deviation (on deproteinized grass, only 1 culture). Variables and parameters are compared graphically in Appendix A, where the results of the statistical analyses are also presented.

Feed Substrate		Wheat Bran	Rapeseed Cake	Brewers’ Spent Grain	Deproteinized Grass
Experimental variables					
*X_DW,_* _14_	mg	0.99 ± 0.07	0.83 ± 0.04	0.85 ± 0.05	0.60 ± 0.03
*t_p_*	days	65	93	n.d.	n.d.
Survival rate	%	98 ± 1	96 ± 2	91 ± 4	0–20
*μ*	day^−1^	0.10 ± 0.00	0.06 ± 0.00	0.02 ± 0.00	0.01 ± 0.01
*a_Day_* _19_	day^−1^	0.34 ± 0.00	0.37 ± 0.05	0.17 ± 0.01	0.16 ± 0.04
SCE	-	0.14 ± 0.01	0.12 ± 0.00	0.02 ± 0.00	0.00
Model parameters					
*X_DW,max_*	mg	78 ± 0	56 ± 3	7 ± 6	n.d.
*μ_max_*	day^−1^	0.11 ± 0.00	0.07 ± 0.00	0.03 ± 0.01	0.02
*Y*	-	1.08 ± 0.08	0.36 ± 0.32	0.79 ± 0.14	0.39
*m*	day^−1^	0.03 ± 0.00	0.05 ± 0.01	0.03 ± 0.01	0.02
*b*	-	0.53 ± 0.1	0.43 ± 0.1	0.51 ± 0.1	0.53
NGE*_avg_	-	0.40 ± 0.01	0.37 ± 0.03	0.17 ± 0.02	0.16
NGE*_DW,avg_	-	0.32 ± 0.01	0.28 ± 0.02	0.13 ± 0.02	0.12
Larval composition					
*X_WW_*	mg	185 ± 9	129 ± 7	12 ± 4	6
*δ_DW_*	% WW	37 ± 0	42 ± 1	34 ± 1	23
*δ_C_*	% DW	59 ± 3	59 ± 3	60 ± 15	53
*δ_N_*	%DW	8.3 ± 0.3	8.6 ± 0.7	12.1 ± 2.4	10.3
*δ_protein_*	% DW	39 ± 2	40 ± 3	56 ± 11	48
*δ_lipid_*	% DW	29 ± 1	26 ± 1	n.d.	n.d.
*δ_ash_*	% DW	2 ± 0	2 ± 0	n.d.	n.d.

**Table 3 animals-15-00233-t003:** Experimental variables, model parameters, and tissue composition at harvest of the BSF larvae reared on different feed substrates. Dry weight of larvae, age 7 days (first measurement), *X_DW_*_7_. Time until >5% of the BSF larvae developed into prepupae, *t_p_*. Specific growth rates, *μ* (determined from Appendix A). Substrate conversion efficiency, SCE (Equation (1)). Maximal DW of larvae, *X_DW,max_,* and maximum specific growth rate, *μ_max_* (Equation (2)). Maximal feed assimilation rate, *a_max_* (Equation (8)). Cost of growth, *Y*, maintenance coefficient, *m*, and allometric coefficient, *b* (Equation (6)). Average carbon net growth efficiency, NGE*_avg_ (Equation (10)), and average net growth efficiency based on mass, NGE*_DW,avg_. Wet weight, *X_WW_*. DW fraction, *δ_DW_*. Contents of carbon, *δ_C_*, nitrogen, *δ_N_*, proteins, *δ_protein_*, lipids, *δ_lipid_*, and inorganic components (ash), *δ_ash_* in the BSF larvae at time of harvest. Measurements represent averages of 5 replicate cultures ± standard deviation. Variables and parameters are compared graphically in Appendix A, where the results of the statistical analyses are also presented.

Feed Substrate		Chicken Feed	Rapeseed Cake	Brewers’ Spent Grain	Deproteinized Grass	Biopulp
Experimental variables						
*X_DW,_* _7_	mg	0.67 ± 0.10	0.92 ± 0.10	0.85 ± 0.07	0.73 ± 0.08	0.82 ± 0.03
*t_p_*	days	23	23	25	25	25
Survival rate	%	99 ± 1	97 ± 2	99 ± 1	98 ± 2	96 ± 2
*μ*	day^−1^	0.85 ± 0.03	0.76 ± 0.04	0.56 ± 0.03	0.65 ± 0.02	0.73 ± 0.04
SCE	-	0.35 ± 0.01	0.29 ± 0.01	0.12 ± 0.01	0.14 ± 0.02	0.35 ± 0.02
Model parameters						
*X_DW_._max_*	mg	113 ± 11	90 ± 6	28 ± 1	35 ± 2	96 ± 1
*μ_max_*	day^−1^	0.77 ± 0.03	0.77 ± 0.07	0.50 ± 0.05	0.59 ± 0.03	0.63 ± 0.01
*a_max_*	day^−1^	1.04 ± 0.08	0.95 ± 0.07	0.96 ± 0.06	1.16 ± 0.03	0.81 ± 0.03
*Y*	-	0.18 ± 0.06	0.12 ± 0.03	0.13 ± 0.11	0.73 ± 0.05	0.19 ± 0.04
*m*	day^−1^	0.12 ± 0.01	0.09 ± 0.01	0.21 ± 0.03	0.13 ± 0.01	0.07 ± 0.00
^1^ NGE*_avg_	-	0.50 ± 0.04	0.56 ± 0.01	0.33 ± 0.02	0.34 ± 0.01	0.56 ± 0.01
NGE*_DW,avg_	-	0.43 ± 0.03	0.47 ± 0.01	0.28 ± 0.02	0.35 ± 0.01	0.46 ± 0.01
Larval composition						
*X_WW_*	mg	354 ± 24	251 ± 21	115 ± 11	118 ± 17	283 ± 25
*δ_DW_*	% WW	32 ± 0	33 ± 1	28 ± 1	26 ± 2	35 ± 2
*δ_C_*	% DW	55 ± 2	54 ± 2	55 ± 3	44 ± 1	55 ± 2
*δ_N_*	%DW	6.3 ± 0.4	7.2 ± 0.1	7.7 ± 0.3	7.2 ± 0.2	6.1 ± 0.3
*δ_protein_*	% DW	30 ± 2	34 ± 1	36 ± 1	34 ± 1	29 ± 1
*δ_lipid_*	% DW	20 ± 2	28 ± 1	26 ± 1	13 ± 2	40 ± 2
*δ_ash_*	% DW	12 ± 0	9 ± 1	6 ± 1	20 ± 2	11 ± 1

^1^ NGE*_avg_ determined on the second last day of the experiments.

## Data Availability

The original data presented in the study are openly available in Mendeley Data at https://data.mendeley.com/datasets/mfxvk6pwx4 (accessed on 15 December 2024).

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
