# Peer review of "Metabolic Performance of Mealworms and Black Soldier Fly Larvae Reared on Food and Agricultural Waste and By-Products"

_animals, 2025, doi:10.3390/ani15020233_

Round 1
Reviewer 1 Report (Previous Reviewer 2)
Comments and Suggestions for Authors
General comments
The goal of this study is to quantify efficiency of growth of two insect species on several organic residual materials that differ in nutritional quality. This is an understudied topic in the quickly growing literature on these two insect species and as such valuable to be addressed. Measurements were made of larval survival, growth and respiration, the decrease in dry matter of the feed substrate and larval body composition.
Having reviewed a previous version of this mansucript the revised version has been considerably improved. However, a major issue remaining is the reference to the mass balance approach. In response to previous comments, two new supplementary tables (S1, S2) have been included that present the basic numbers for feed substrate mass decrease in dry matter (DM) units and the body weight gain in DM units that allow the calculation of SCE. However, this information does not suffice to construct a complete mass balance. Arguments are detailed below, referring to the respective line numbers in the manuscript.
Specific comments
L. 74: Unclear what is meant by ‘decreased’: relative to what reference value? Decreased by what/how?
L. 94: Correct phrasing’…into to ….during across…’.
Life history of larvae is poorly phrased: the larval phase is a component of an animal’s life history; suggest to rephrase as ‘the larval stage’.
L. 95: NGEavg does not represent a carbon balance of the larvae; this statement is incorrect. For transparency, specify the carbon balance equation and indicate which terms in the equation have been quantified in this paper and which are entered into the equation based on previous studies.
L. 216: Mass balance approach: See comment on L. 95. In any case, an estimate of the total amount of dry matter respired as carbon dioxide by the larvae should be included; this will result in a fraction of dry matter that remains unexplained according to the equation (all terms in DM units):
[Initial feed substrate mass] + [initial larval body mass] = [Final feed substrate mass] + [Larval body mass at end of experiment] + [Respiration by larvae and by microbes].
In this equation microbial respiration is an unknown and it is argued in the manuscript that this is the main process causing discrepancies between SCE and NGEDW,avg. However, no definitive conclusion can be drawn on the contribution of microbial respiration to the unexplained fraction in the mass balance. It should be noted that the measurement of respiration was performed using a closed-vessel respiration chamber in which the larvae had been enclosed without feed substrate; this leaves the possibility open that actual respiration intensity occurring when larvae feed inside the feed substrate was different from the values measured experimentally and may therefore have led to incorrect (under-)estimation of respiration intensity. This limitation that is inherent to the methods used (that do not allow a complete mass balance to be constructed) needs to be addressed in the Discussion.
L. 456-458 and L. 483-486: The suggestion that microbial respiration was contributing more in the dry feed substrates on which mealworms were reared than in the wet feed substate of the BSF larvae is awkward. Provide references that support this suggestion. In my opinion the reverse is more likely: microbial growth is stimulated by higher substrate water contents.
Author Response
We have attached our point-by-point response to Reviewer 1.

Reviewer 2 Report (New Reviewer)
Comments and Suggestions for Authors
The manuscript presents an informative comparative study on the growth performance and feed efficiency of mealworms and black soldier fly larvae (BSF) when reared on diverse feed substrates. The methodology is robust, incorporating detailed metrics for growth, CO2 production, and feed assimilation. The findings emphasize the metabolic and feed efficiency advantages of black soldier fly larvae while highlighting the unique metabolic traits of mealworms. Minor grammatical refinements and phrase restructuring could improve readability. Additionally, consider further elaborating on how the metabolic traits influence broader ecological and industrial applications. Overall, the manuscript is well-structured and contributes valuable insights to the field of sustainable insect farming.
Major comments:
Use consistent terminology (e.g., "feed efficiency" versus "feed efficiencies").
Avoid overly long sentences by breaking them into shorter, more digestible parts for improved readability.
The proximate composition analysis needs to be further defined with more details. I am confused about the fat analysis with NMR. According to the AOAC method, crude fat content could be determined by the measurement of ether extract which does not require NMR instrument.
Minor comments:
L12: “two of the industrially most used insects” Suggestion: “two of the most widely used industrial insects”
L52: Add hyphenation to make ("species-dependent")
L57: “which gradually are” Suggestion: “which are gradually”
L65: “have been reared successfully” Suggestion: “have been successfully reared”
L95: Replaces "a carbon balance" with "the carbon balance" for specificity.
L119: “every third day” Suggestion: “every three days”
L171: change “euthanized and” to “euthanized, and” Add a missing comma before "and stored"
L209: Original: "Then, 2 mL of methanol and 1 mL of chloroform were added, and the mixture was left on a shaker for 20 min followed by the addition of 1 mL of chloroform and one more minute of shaking."
Suggested: "Next, 2 mL of methanol and 1 mL of chloroform were added, and the mixture was shaken for 20 minutes. An additional 1 mL of chloroform was then added, followed by one minute of shaking."
Breaks the sentence for better readability.
L278: change “presumable” to “presumably”
L333: change “ranged from 0.02-0.05” to “ranged from 0.02 to 0.05”
L477: “main factor” Suggested: “primary determinant”
Author Response
We have attached our point-by-point response to Reviewer 2.

Reviewer 3 Report (New Reviewer)
Comments and Suggestions for Authors
The importance of proteins of insects origin attracted considerable and increasing interest as a feed component that provides higher local independence in the supply of feed proteins with higher sustainability as its production and use lead to lower carbon footprint and less negative impact on the environment and biodiversity compared with plant protein. Since the European Union law allowed to use insects in farm animal feed, a lot of research focused of the usage of different insects meal for usefulness as feed, especially in poultry. Only a few studies focus on assessing the growth and metabolic performance and feed efficiency of mealworms and black soldier fly larvaes fed with mixtures of different origin. Taking the above into account, the presented research results have a significant contribution to the development of an effective method of rearing insects larvae used to meet the growing demand for protein of insects larvae meals. The article is well written and has high scientific value. However, I have minior comment regarding the use of wordings:
line 27: I suggested to use "better" insted of "bigger" because word "bigger" is used to describe the size of something. To discuss the scientific results, usually we use: lower, higher, better...
Author Response
Thank you for the positive evaluation. We have replaced ‘bigger’ with ‘stronger’.
Round 2
Reviewer 1 Report (Previous Reviewer 2)
Comments and Suggestions for Authors
The rebuttal and the corresponding revisions made in response to the comments on the previous version of this manuscript are adequate and have rendered the data presentation as well as their interpretation fully transparent.
Below a few improvements in phrasing are suggested and a number of typos need to be corrected; these are listed below:
Figure 5, typo in legend: it should read ‘B. BSF ...’.
Figure S2: in figure legend, p = 0.00 is unclear; state a discrete p-value, or state ‘p < [discrete maximal value]’.
Table S1: typo in header of fifth column; should read ‘Brewers’ spent grain’.
Table S1: Table header: suggest to change to ‘Start and end masses….’ since the numbers presented do not add up to a balance; carbon balances in this revised version are presented in Tables S2 and S4. In addition, typos occur in this table header:
‘a harvest’ should read ‘at harvest’;
Third line: delete ‘to’
‘Substate’ should read ‘Substrate’.
Table S2: Table header: suggest to change to ‘…changes in amount of carbon’; the amounts of carbon make up the balance, not carbon contents.
Typo in header of fifth column; should read ‘Brewers’ spent grain’.
Table S2: typo in column 1, second underlined entry: should read ‘mealworm’
Table S2: typos in footnotes 6, 7, 8 and 9: the ‘2’ in CO2 should appear as subscript.
Table S2, footnote 7: in subscript with C it should read ‘survivors’.
Table S2, footnote 9: typo in subscript, should read ‘balance’.
Figure S4: in figure legend, p = 0.00 is unclear; state a discrete p-value, or state ‘p < [discrete maximal value]’.
Figure S4F: remove ‘, SCE’ along Y-axis.
Table S3: see comments on text of header of Table S1.
Table S4: see comments on text of header of Table S2.
Table S4: see the typos noted for Table S2.
Tabel S5: should read ‘Table S5’ and ‘og’ should read ‘or’(?)
Author Response
We have attached our point-by-point response. Thank you for reading the manuscript with care.

This manuscript is a resubmission of an earlier submission. The following is a list of the peer review reports and author responses from that submission.
Round 1
Reviewer 1 Report
Comments and Suggestions for Authors
R12: 2 needs to be in full text
R18: "it is" replace by "the"
R18: 2 needs to be in full text
R20: 2 needs to be in full text
R72 remove "dry weight"
R94 ultimately impacting is in another font
R111: 10-12 days seems long to me. Normally 7-9 days
R131: Why did you use two different breweries?
R134: In my opinion, this produces drier and much wetter substrates compared to each other. I think it would be better almost fully saturate the substrates with water.
Frooninckx, L., Broeckx, L., Goossens, S. et al. Optimizing substrate moisture content for enhanced larval survival and growth performance in Hermetia illucens: exploring novel approaches. Discov Anim 1, 7 (2024). https://doi.org/10.1007/s44338-024-00005-2
R142: It is a lot of DW per individual.
Deruytter, D., Rumbos, C.I., Adamaki-Sotiraki, C., Tournier, L., Ageorges, V., Coudron, C.L., Yakti, W., Ulrichs, C., Spranghers, T., Berrens, S., Van Peer, M., Bellezza Oddon, S., Biasato, I., Resconi, A., Paris, N., Hotte, N., Hénault-Ethier, L., Gasco, L., & Athanassiou, C.G. (2024). Make it a standard? The creation and variability assessment of a consensus standard protocol for Tenebrio molitor larvae feeding trials. Journal of Insects as Food and Feed (published online ahead of print 2024). https://doi.org/10.1163/23524588-00001347
R148-150: It is a lot of DW per individual.
Deruytter, D., Gasco, L., Yakti, W., Katz, H., Coudron, C.L., Gligorescu, A., Frooninckx, L., Noyens, I., Meneguz, M., Grosso, F., Bellezza Oddon, S., Biasato, I., Mielenz, M., Veldkamp, T., Van Loon, J.J.A., Spranghers, T., Vandenberg, G.W., Oonincx, D.G.A.B., & Bosch, G. (2023). Standardising black soldier fly larvae feeding experiments: an initial protocol and variability estimates. Journal of Insects as Food and Feed, 10(10), 1685-1696. https://doi.org/10.1163/23524588-20230008
R150: add the amount of DW that was added the second time
R154-155: express why you chose for this termination criterion and not on a certain day or weight.
R163: This is not the most accurate sensor. Are you sure that there is no variation due to this sensor?
R181: 6.25 is an old factor. Use 5.33 instead for mealworms and 4.67 for BSFL.
Boulos S, Tännler A, Nyström L. Nitrogen-to-Protein Conversion Factors for Edible Insects on the Swiss Market: T. molitor, A. domesticus, and L. migratoria. Front Nutr. 2020 Jul 10;7:89. doi: 10.3389/fnut.2020.00089. PMID: 32754611; PMCID: PMC7366252.
Janssen, R.H., Vincken, J.-P., Van den Broek, L.A.M., Fogliano, V. and Lakemond, C.M.M., 2017. Nitrogen-to-protein conversion factors for three edible insects: Tenebrio molitor, Alphitobius diaperinus, and Hermetia illucens. Journal of Agricultural and Food Chemistry 65: 2275-2278.
R194: Mention how you determined the larval weight. 1 larvae each time, x larvae or all larvae
Equation 6: Divided by X --> not 1-b but b-1
Check also all the tables and graphs where you used this formula
R258-259: If you use another equation 2 with a inflection point, it would be easier to take the LN. Up to this inflection point there is exponential growth and you have a linear phase after taking the logarithm. This will make it easier to compare. Mention also R²-values
In figure 1: eq 7 is eq 6
In figure 1: wrong symbol for extinction of individual larval cultures
In table 2: mention X0
In table 2: 43% difference and 93 days is not acceptable for a mealworm producer. Express this more in your text and discussion
In table 2: 66 (% prot in DW) is less than 12.1x6.25. Recalculate with the more recent factor and if there is a difference mention why
R274: add r²-values in the text or on the figure
R289: Weight in stead of carbon content? Otherwise explain more
R295: Unclear why 1mmol carbon is equal to 25 mg DW. Explain
R316-317: I think there is a significant difference between the different feeds.
Do a statistical comparison: ANOVA but I think a non-parametric comparison will be better. Mention also the R²-values.
In table 3: recalculate with new N to protein values and mention in the text if this calculated protein value would be different from the conversion factor
In table 3: More Nitrogen in rapeseed cake, but less protein than larvae reared on chicken feed.
R363: it seems to me that their are significant differences between the larvae reared on different feed.
Do a statistical comparison: ANOVA but I think a non-parametric comparison will be better.
R372: a bit strange to mention in your abstract that only 2 out of 4 and 3 out of 5 went well.
R380: 2 should be two
R384-385: Mention r²-value and/or p-value from your regression analysis
R414: 2 should be two
R419: The molt could also be an important factor why your survival is so low. Elaborate on this and why or why this had no effect on your experiment.
R440: 2 should be two
R456: 3 should be three
R473: 2 should be two
R495: 4 should be four

Author Response
Please find our point-to-point response in the attached file.

Reviewer 2 Report
Comments and Suggestions for Authors
General comments
The paper presents experimental results on two insect species that are mass-reared for conversion of organic residues into biomass production form which insect proteins, fat and chitin can be extracted. The goal of this study is to quantify efficiency of growth of the two species on several organic residual materials that differ in nutritional quality. This is an understudied topic in the quickly growing literature on these two insect species and as such valuable to be addressed. Measurements were made of larval survival, growth and respiration. The assessment of growth efficiency is expressed in carbon units and is indirect since the fraction of feed assimilated is estimated based on curve fitting and previous data for maintenance metabolism. Although the limitations of this approach and its possible effects on the accuracy of estimation of growth efficiency is addressed in the Discussion, this approach is fundamentally flawed. In my opinion the only sound experimental approach is a mass balance approach; this approach is mentioned in Materials and methods, however, no corresponding data are presented. A complete mass balance is in my opinion crucial to validate if the loss of carbon as carbon dioxide resulting from larval respiration is what was actually respired by larvae feeding in the substrate. For a complete mass balance data on the carbon content of the feed residue at the end of the experimental period are essential. If these data are lacking the accuracy of the quantification of growth efficiency cannot be checked. In addition, whereas there were five experimental replicates, no measure of variability is presented in growth efficiency precluding statistically solid comparisons between the two species and between data in the lierature and the experimental data in the manuscript. Whereas I appreciate the topic of the study as interesting and the approach to estimate the fraction of feed assimilated as a potential solution to estimate growth efficiency of insects living inside their feed substrate for which it is not feasible to quantify the amount of feed ingested and the fraction of feed ending up as faecal material, the lack of a carbon mass balance and the assumptions made on maintenance metabolism and on variables necessary to allow comparisons with literature data make it unconvincing, in particular from an animal physiological perspective.
Specific comments
L. 14, 16, 28, 30, 577, 578: advantageous / disadvantageous; advantages/disadvantages: specify in which respect(s).
L. 27; L. 380: ‘pickiest’ is not applicable since this term refers to a choice situation whereas the tests were performed in a no-choice situation. Suggest to rephrase as: ‘Mealworms grew well on …’
L. 47-48: ‘feed efficiency’ is an unclear term and although used by some authors, it is suggested to rename it throughout the manuscript into ‘feed conversion efficiency’.
L. 71-72: See comment on L. 47-48: ‘substrate conversion efficiency (SCE)’: in one of the experimental papers on BSF larvae cited (20), conversion efficiency was calculated as insect biomass (DM) collected divided by the amount diet provided (DM), therefore calculated differently than SCE and consequently citations of the numbers presented here are not values of SCE. Citations are therefore incorrect. Ref. 21 is a review and in Table 1 lists values of ‘bioconversion rate’, not of SCE.
Furthermore, the calculation formula for SCE has no physiological relevance since the reduction in dry weight of the feed mass is not a measure of the amount of feed ingested; the residue remaining at the end of the feeding experiment is a mixture of faeces and unconsumed feed in unknown proportions. Although it is appreciated that this is an inherent limitation imposed by the lifestyle of the two species, i.e. they live inside the feed substrate and mix the faeces they produce with feed and may unavoidably perform coprophagy, using the mass of the residue left at the end of the feeding experiment does not provide insight in conversion efficiency from a physiological perspective.
L. 91: should read ‘..of what corresponds to..’.; or this part of the sentence could be deleted.
L. 115: Specify the rearing substrate on which the 7-day old BSF larvae had been reared since the specific pre-rearing substrate used may have influenced the results obtained for the five experimental substrates. In order to allow replication of the experiments, this information should be provided.
L. 126: blending: how was this done? What were the resulting particle sizes?
L. 134: adjusted moisture content: adjusted how and with what?
L. 136: Specify the accreditation in terms of ISO certificates.
L. 141: Insert spaces between number and unit throughout the manuscript.
L. 144-145: One replicate for DW determination is not sufficient; in addition, it changes larval density over time, possibly leading to differences in DW compared to the experimental replicates.
L. 152-153: this procedure is better than the one applied for yellow mealworms commented on above (L. 144-145). Yet also here, for each timepoint there is just one replicate, one group of larvae. However, in the graphs a measure of variation is depicted: please clarify how this variation has been calculated.
L. 156: euthanized: how?
L. 170-171: this describes how ash was produced from larval samples; this is not sufficient to determine ash content; specify the follow-up steps.
L. 181: The use of the conversion factor 6.25 is not accurate; see Jansen et al. (2017) Journal of Agricultural and Food Chemistry 65, 2275−2278. Since the deviations from the value 6.25 differ between yellow mealworm and BSF larvae, these corrections should lead to corrections in the values presented in the manuscript.
L. 191: Mass balance approach: see General comments; in the manuscript there are no data presented to substantiate this; no mass balances are presented, neither in the main text nor in Suppl. Materials.
L. 218: qO2 should read qCO2
L. 447: ‘quit’ should read quite
L. 459: incomplete sentence; word(s) seem to be missing.
L. 469: ‘studied’ should read ‘study’.
L. 534-535: See comment on L. 181: rather than concluding that a large overestimation may have occurred, it is warranted to use the conversion factors as determined by Jansen et al. (2017) for both BSF and mealworms. The values for larval body protein contents in Tables 2 and 3 should be corrected accordingly.
L. 535: specie should read species
L. 549: Here it seems appropriate to point out that the base line substrates were (most likely; see comment on L. 115) also the substrates of the colonies from which the experimental larvae originated and that therefore the larvae have been selected for high performance on these substrates during many generations.
L. 556: Reference numbers [49-59] incorrect.
L. 560: word missing.
L. 577, 578: advantageous / disadvantageous: specify in what respect.
Author Response

(The authors gave the same response as above.)

Reviewer 3 Report
Comments and Suggestions for Authors
It would be helpful if the authors consistently clarified units and terminology throughout the manuscript. For instance, terms such as "NGE*" and "SCE" should be explicitly defined when first introduced to aid reader comprehension.
The paper would benefit from further explanation regarding the statistical analysis methods used. Details about why specific statistical tests were chosen and any assumptions made would enhance the credibility of the findings. Furthermore, reporting effect sizes alongside p-values could provide a more comprehensive understanding of the results.
The paper contains various comparisons between mealworms and BSF larvae. The clarity of these comparisons could be improved by incorporating additional graphs or tables that directly compare these metrics side-by-side, enhancing the reader's ability to observe differences visually.
The conclusions section could be expanded to emphasize the practical applications of the findings in industrial settings. Discussing potential implications for insect farming efficiency based on the results, especially in sustainable agriculture or waste management, would provide additional value for industry professionals interested in applying these findings.
Comments on the Quality of English LanguageNo.
Author Response

(The authors gave the same response as above.)
